# Impact of lifestyle behaviors on the development of lifestyle diseases: A retrospective cohort study

Takafumi Okawa[1,2], Hikaru Negishi[2], Yuki Aoki[1‡], Mitsuo Uchida[1‡], Yumi Sato[3‡], Mai Ishikawa[3], Rie Matsui[3], Kaori Hotta[3], Takayuki Saitoh[2*]

**1** Department of Mathematics and Data science, Graduate School of Medicine, Gunma University, Gunma, Japan, **2** Department of Laboratory Sciences, Graduate school of Health Sciences, Gunma University, Gunma, Japan, **3** Department of Nursing, Graduate School of Health Sciences, Gunma University, Gunma, Japan

☯ These authors contributed equally to this work.
‡ These authors also contributed equally to this work.
* tsaitoh@gunma-u.ac.jp

## Abstract

### Background

Questionnaires are used to collect data on lifestyle behaviors during specific health checkups; however, the results cannot conclusively determine whether the behaviors influence the onset of lifestyle diseases. By analyzing data from a retrospective cohort, this study aimed to determine the specific lifestyle behaviors that most strongly contribute to the onset of lifestyle diseases, such as metabolic syndrome, hypertension, diabetes, and dyslipidemia.

### Methods

We administrated the data of 924,932 individuals insured under Gunma Prefecture's National Health Insurance who underwent specific health checkups between 2011 and 2016. A logistic regression analysis was conducted to assess the association between the responses to 10 lifestyle questions and the future onset of lifestyle diseases.

### Results

We examined 47,803 individuals who were not identified with lifestyle disorders at the initial checkup. In this study, weight gain of ≥10 kg since the age of 20 years showed the strongest association with MetS (OR: 2.01; 95% CI, 1.79–2.25). Additionally, smoking and weight gain were identified as common risk factors for MetS, hypertension, and dyslipidemia. The results revealed that lifestyle behaviors are longitudinally associated with the onset of lifestyle diseases.

---

**Data availability statement:** The Ethics Committee for Research Involving Humans of the Gunma University Faculty of Medical applies the restriction for public data sharing due to ethical and legal restrictions of the annual health check-up data containing sensitive information. De-identified data may be available upon reasonable request and subject to approval by the Ethics Committee. Requests for data access should be directed to the Office of the Ethics Committee for Research Involving Humans, Advanced Medical Development Center, Gunma University (Email: hitotaisho-ciru@ml.gunma-u.ac.jp).

**Funding:** TS received funding from JST (Japan Science and Technology Agency), Grant Number JPMJPF2301.

**Competing interests:** The authors have declared that no competing interests exist.

## Conclusion

The use of self-administered questionnaires to assess lifestyle behaviors can effectively predict future health risks.

## Introduction

Aging has recently become a prevalent issue in developed countries. Japan, particularly, is one of the leading super-aged societies worldwide. The increase in non-communicable diseases (NCDs) with the aging population has led to a rise in national healthcare expenditures, making cost containment a critical public health priority [1–3]. Specific health checkups (SHC) have emerged as initiatives aimed at curbing the rising national healthcare costs in Japan's aging society [4].

The SHC program provides health screening and promotion services tailored to insured individuals aged 40–74 years and their dependents. Its primary goal is to prevent bedridden conditions in older adults through the early detection and intervention of metabolic syndrome (MetS). The SHC is a unique initiative in Japan, established by the Ministry of Health, Labour and Welfare (MHLW) in 2008 under the "Act on Assurance of Medical Care for Elderly People" [4]. Currently, approximately 25.4 million beneficiaries are covered by the municipal national health insurance, representing a significant effort to promote the health of older adults at the municipal level [5]. Notably, the MHLW's "Guide to Creating a Data Health Plan, 3rd Revised Edition" emphasizes that utilizing SHC data can facilitate early detection of lifestyle diseases, thereby contributing to the reduction of national medical care expenditures [6]. To date, health promotion initiatives, utilizing the SHC data, are currently being implemented in Shizuoka and Osaka Prefectures [7,8].

The SHC utilizes the use of anthropometric measurements, laboratory values, and self-administered questionnaires designed to assess lifestyle behaviors. These questionnaires conform to the "standard questionnaire" established by the MHLW [9]. However, the evidence derived from the questionnaire results was not consistent in predicting the onset of lifestyle diseases [10–14]. For example, some studies highlight a correlation between skipping breakfast and the onset of diabetes [15,16], whereas others refute this connection [17]. This inconsistency has resulted in the inadequate utilization of data for formulating health management policies.

The discrepancy in conclusions stems from several factors, such as the inclusion of individuals with positive results, which could affect the accurate assessment of the effects of lifestyle behaviors [18], inadequate long-term evaluations [17], and insufficient consideration of the interaction effects of lifestyle behaviors on the onset of lifestyle diseases. Addressing these issues through the use of adequate sample size and the conduct of a longitudinal analysis could lead to more consistent conclusions regarding this relationship.

Using SHC data collected over 10 years from the National Health Insurance (NHI) database, our retrospective cohort study aimed to investigate lifestyle behaviors that

are genuinely associated with the onset of lifestyle diseases such as MetS and its components such as hypertension, diabetes, and dyslipidemia.

## Materials and methods

### Database

We used SHC data from the Gunma Prefecture's NHI database, covering a 10-year period from 2011 to 2020. The Gunma Prefecture has a population of approximately 2 million, with about 20% of individuals' insurance data updated annually in the NHI database.

### Study population

The selection process for the study population is outlined in Fig 1. The study included 924,932 total counts of individuals insured under Gunma Prefecture's NHI who underwent SHC between 2011 and 2016 (150,610 in 2011, 154,579 in 2012, 156,454 in 2013, 156,531 in 2014, 155,690 in 2015, and 151,068 in 2016). The onset of lifestyle diseases was assessed 4 years after the initial checkup, based on previous research [11]. To ensure the suitability of the study samples, individuals with missing health checkup data 4 years post-initial checkup (N = 447,511); whose presence or absence of lifestyle diseases were not determined (N = 3,470); who were identified with MetS, diabetes, hypertension, or dyslipidemia at the initial checkup (N = 363,453); who reported a history of stroke, heart disease, or chronic kidney failure in the self-administered

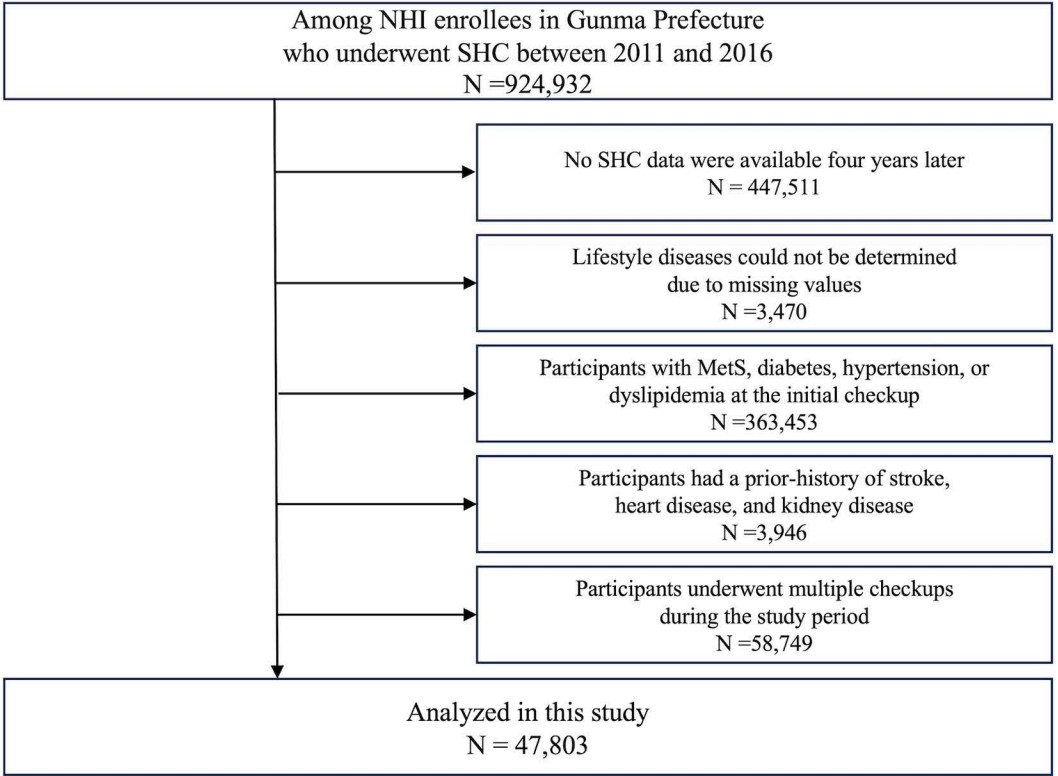

**Fig 1. Study population.** The selection process for the study population. Abbreviations: NHI, National Health Insurance; SHC, Specific health check-ups; MetS, metabolic syndrome.

questionnaire during the initial health checkup (N = 3,946); and who underwent multiple checkups during the study period (N = 58,749) were excluded. Hence, 47,803 participants were included in the final analysis.

## Variables

The following data were extracted from the SHC records: personal information (sex, date of birth, and date of SHC); anthropometric measurements (body mass index [BMI] and waist circumference [WC]); laboratory values (systolic blood pressure [SBP], diastolic blood pressure [DBP], hemoglobin A1c [HbA1c], triglyceride [TG], high-density lipoprotein cholesterol [HDL-C], low-density lipoprotein cholesterol [LDL-C], aspartate aminotransferase [AST], alanine aminotransferase [ALT], and γ-glutamyl transpeptidase [γ-GTP] levels). The results of self-administered questionnaire survey on lifestyle behaviors (smoking status, weight change, exercise habits, eating habits, drinking habits, and sleep habits), medications used (for hypertension, dyslipidemia, and diabetes), and disease history (stroke, heart disease, and kidney disease) were also extracted. Age, BMI, and laboratory values were categorized according to the medical guidelines, as detailed in S1 Table. Data on lifestyle behaviors at the initial checkup were extracted from the results of the "standard questionnaire" provided by the MHLW. From this questionnaire, 11 questions that have consistently provided valid responses over the years were selected (as shown in S2 Table). The presence or absence of lifestyle behavior risks is also presented in S2 Table. With regard to drinking habits, the presence or absence of risk was determined based on the frequency of drinking (Question 9) and amount of alcohol consumed (Question 10), following the criteria used in previous studies [19]. Individuals who reported drinking "every day" or "sometimes" and consuming "180 mL or more" of alcohol were classified as having a risk. Conversely, those who drank "hardly ever" or consumed "less than 180 mL" were classified as not having a risk. A serving of 180 mL of refined saké (rice wine) is equivalent to a medium bottle of beer (500 mL), 110 mL of shochu (25% alcohol content), a double shot of whiskey (60 mL), or two glasses of wine (240 mL).

In this study, we selected MetS and its components—hypertension, diabetes, and dyslipidemia—to investigate their association with lifestyle behaviors. MetS was defined according to the criteria of the National Health and Nutrition Survey (NHNS) [20], which requires the presence of visceral fat accumulation (WC ≥ 85 cm in men and ≥ 90 cm in women) and two or more of the following criteria: 1) an SBP > 130 mmHg, DBP of >85 mmHg, or the use of antihypertensive drugs; 2) an HDL-C level <40 mg/dL or the use of antihyperlipidemic drugs; and 3) an HbA1c level ≥6.0% or the use of antidiabetic drugs. Hypertension was defined as an SBP ≥ 140 mmHg, a DBP ≥ 90 mmHg, or the use of antihypertensive drugs. Diabetes was defined as an HbA1c level ≥6.5% or the use of antidiabetic drugs. Dyslipidemia was defined as a TG level ≥150 mg/dL, an HDL-C level <40 mg/dL, an LDL-C level ≥140 mg/dL, or the use of antihyperlipidemic drugs. Individuals who did not meet any of the above criteria were considered healthy.

## Statistical analysis

Chi-square tests were performed to compare the participant characteristics. The odds ratios (OR) and corresponding 95% confidence intervals (CI) for the onset of lifestyle diseases were calculated using a logistic regression model. Initially, a simple regression model was employed, using 10 lifestyle behaviors as independent variables and MetS, hypertension, diabetes, and dyslipidemia as dependent variables. Two adjusted models were subsequently employed. Model 1 included the 10 lifestyle behaviors as independent variables and was adjusted for sex and age as covariates. Model 2 was adjusted for Model 1 and addition of BMI and laboratory tests as covariates. Correlation coefficients and variance inflation factors (VIFs) were calculated for each variable to assess multicollinearity among the independent variables, confirming that all VIF values were below 5. All data were analyzed using Python 3.10 and a p value of <0.05 was considered significant.

## Ethical considerations

The study was approved by the Ethics Committee for Research Involving Humans of the Gunma University Faculty of Medical, Gunma, Japan (HS2023−137). This study involved a secondary analysis of data from the existing Kokuho database. As the data had already been anonymized at the time of acquisition, informed consent was not required.

 

## Results

The characteristics of participants who developed lifestyle diseases and those who did not are presented in Table 1. The BMI and laboratory values were compared between the participants who developed lifestyle diseases and those who did not (S3 Table). The prevalence of MetS 4 years after the initial health checkup was 4.8%. Individuals with MetS were more likely to engage in risky lifestyle behaviors, such as smoking, weight gain ≥10 kg since the age of 20 years, decreased physical activity, slow walking, fast eating, eating before bedtime, skipping breakfast, and problematic drinking habits. Conversely, they were less likely to exhibit adopting poor sleeping habits. The prevalence of hypertension 4 years after the initial health checkup was 26.1%. Individuals with hypertension were more likely to exhibit risky lifestyle behaviors such as smoking, weight change ≥10 kg since the age of 20 years, eating before bedtime, and problematic drinking habits. Conversely, they were less likely to exhibit risky behaviors, including the lack of regular exercise, decreased physical activity, skipping breakfast, and adopting poor sleeping habits. The prevalence of diabetes 4 years after the initial health checkup was 2.5%. Individuals with diabetes were more likely to exhibit risky lifestyle behaviors such as smoking, weight change ≥10 kg since the age of 20 years, and fast eating. Conversely, they were less likely to engage in risk behaviors, such as the lack of regular exercise, skipping breakfast, and adopting poor sleeping habits. The prevalence of dyslipidemia 4 years after the initial health checkup was 33.2%. Individuals with dyslipidemia were more likely to exhibit risky lifestyle behaviors, such as smoking, weight change ≥10 kg since the age of 20 years, the lack of regular exercise, decreased physical activity, fast eating, skipping breakfast, and adopting poor sleeping habits. Conversely, they were less likely to exhibit eating behaviors before bedtime and adopt problematic drinking habits.

Table 2 presents the relationship between lifestyle behaviors and lifestyle diseases, determined using a logistic regression analysis. The results of the simple regression model and Model 1 are presented in Table 2. The results of Model 2, adjusted for sex, age, BMI, and laboratory values, are presented below (see Supplementary S1 Fig). MetS demonstrated positive associations with smoking, weight gain ≥10 kg since the age of 20 years, slow walking, and fast eating (Table 2A). The highest OR was observed for weight gain ≥10 kg since the age of 20 years (OR: 2.01, CI: 1.79–2.25). Hypertension was positively associated with smoking, weight gain ≥10 kg since the age of 20 years, the lack of regular exercise, slow walking, and problematic drinking habits (Table 2B). Among these, the highest OR was observed for problematic drinking habits (OR: 1.22, CI: 1.15–1.30). Diabetes was positively associated with fast eating (OR: 1.17, CI: 1.01–1.36). Conversely, skipping breakfast showed a negative relationship with diabetes (OR: 0.70, CI: 0.51–0.95) (Table 2C). Dyslipidemia was positively associated with smoking, weight gain ≥10 kg since the age of 20 years, skipping breakfast, and insufficient sleep. Conversely, drinking habits showed a negative association with dyslipidemia (OR: 0.90, CI: 0.85–0.95) (Table 2D).

## Discussion

This retrospective study analyzed 10 years of accumulated health checkup data to examine lifestyle behaviors associated with the development of MetS and its components—hypertension, diabetes, and dyslipidemia. To our knowledge, this study is the first to assess the impact of lifestyle behaviors on the development of lifestyle diseases using such an extensive dataset, which enhances the statistical power and reliability of the findings. The assessment accurately calculated the positive measurement test results and accounted for the interaction effects of lifestyle behaviors on the onset of lifestyle diseases. New insights were obtained using a two-stage multivariable logistic regression model, where Model 1 and Model 2 were sequentially adjusted to account for potential confounding variables.

First, we identified an association between diabetes and lifestyle behaviors. Smoking [21–23], weight gain of ≥10 kg since the age of 20 years [24,25], the lack of regular exercise [26,27], slow walking [23], fast eating [14,17,23,28,29], and adopting poor sleeping habits [23] have been reported as risk factors for diabetes. Our study showed that fast eating was the sole risk factor in Model 2, while three factors were identified in Model 1 (Table 2C). The remaining two factors, weight gain of ≥10 kg since the age of 20 years and problematic drinking habit, were significantly associated with measurement tests that influence the onset of diabetes and may serve as confounders in this relationship [30,31]. Notably,

**Table 1. Baseline characteristics of individuals who developed and did not develop lifestyle diseases.**

| | | Metabolic syndrome | | | | P value[a] | Hypertension | | | | P value[a] |
|---|---|---|---|---|---|---|---|---|---|---|---|
| | | Undeveloped | | Developed | | | Undeveloped | | Developed | | |
| | | n = 45,507 | | n = 2,296 | | | n = 35,339 | | n = 12,464 | | |
| | | n | (%) | n | (%) | | n | (%) | n | (%) | |
| Attributes | | | | | | | | | | | |
| Sex | | | | | | | | | | | |
| | Men | 17,015 | (37.4) | 1,636 | (71.3) | < 0.001 | 13,189 | (37.3) | 5,462 | (43.8) | < 0.001 |
| | Women | 28,492 | (62.6) | 660 | (28.7) | | 22,150 | (62.7) | 7,002 | (56.2) | |
| Age | | | | | | | | | | | |
| | 40-44 | 3,225 | (7.1) | 110 | (4.8) | < 0.001 | 2,936 | (8.3) | 399 | (3.2) | < 0.001 |
| | 45-49 | 3,203 | (7.0) | 126 | (5.5) | | 2,791 | (7.9) | 538 | (4.3) | |
| | 50-54 | 3,027 | (6.7) | 134 | (5.8) | | 2,544 | (7.2) | 617 | (5.0) | |
| | 55-59 | 3,946 | (8.7) | 213 | (9.3) | | 3,184 | (9.0) | 975 | (7.8) | |
| | 60-64 | 9,203 | (20.2) | 498 | (21.7) | | 6,925 | (19.6) | 2,776 | (22.3) | |
| | 65-69 | 16,194 | (35.6) | 957 | (41.7) | | 11,768 | (33.3) | 5,383 | (43.2) | |
| | ≥ 70 | 6,709 | (14.7) | 258 | (11.2) | | 5,191 | (14.7) | 1,776 | (14.2) | |
| Current Smoking | | | | | | | | | | | |
| | Yes | 6,077 | (13.4) | 524 | (22.8) | < 0.001 | 4,739 | (13.4) | 1,862 | (14.9) | < 0.001 |
| | No | 39,427 | (86.6) | 1,770 | (77.2) | | 30,598 | (86.6) | 10,599 | (85.1) | |
| Weight gain of ≥10 kg since age 20 | | | | | | | | | | | |
| | Yes | 7,087 | (17.3) | 1,162 | (57.2) | < 0.001 | 5,590 | (17.6) | 2,659 | (23.5) | < 0.001 |
| | No | 33,984 | (82.7) | 870 | (42.8) | | 26,218 | (82.4) | 8,636 | (76.5) | |
| Regular exercise | | | | | | | | | | | |
| | Yes | 18,434 | (42.3) | 945 | (44.0) | 0.126 | 14,208 | (41.9) | 5,171 | (43.7) | 0.001 |
| | No | 25,127 | (57.7) | 1,202 | (56.0) | | 19,661 | (58.1) | 6,668 | (56.3) | |
| Physical activity | | | | | | | | | | | |
| | Yes | 21,483 | (52.3) | 1,001 | (49.4) | 0.010 | 16,487 | (51.9) | 5,997 | (53.1) | 0.022 |
| | No | 19,557 | (47.7) | 1,025 | (50.6) | | 15,294 | (48.1) | 5,288 | (46.9) | |
| Walking speed | | | | | | | | | | | |
| | Slow | 18,183 | (44.6) | 994 | (49.1) | < 0.001 | 14,157 | (44.8) | 5,020 | (44.7) | 0.840 |
| | Fast | 22,629 | (55.4) | 1,031 | (50.9) | | 17,445 | (55.2) | 6,215 | (55.3) | |
| Eating speed | | | | | | | | | | | |
| | Slow/Normal | 32,197 | (78.4) | 1,396 | (68.7) | < 0.001 | 24,832 | (78.1) | 8,761 | (77.5) | 0.215 |
| | Fast | 8,867 | (21.6) | 635 | (31.3) | | 6,963 | (21.9) | 2,539 | (22.5) | |
| Eating before bedtime | | | | | | | | | | | |
| | Yes | 6,300 | (15.3) | 416 | (20.5) | < 0.001 | 4,838 | (15.2) | 1,878 | (16.6) | < 0.001 |
| | No | 34,758 | (84.7) | 1,618 | (79.5) | | 26,959 | (84.8) | 9,417 | (83.4) | |
| Skipping breakfast | | | | | | | | | | | |
| | Yes | 3,182 | (7.8) | 197 | (9.7) | 0.002 | 2,546 | (8.0) | 833 | (7.4) | 0.035 |
| | No | 37,869 | (92.2) | 1,831 | (90.3) | | 29,246 | (92.0) | 10,454 | (92.6) | |
| Drinking habits | | | | | | | | | | | |
| | Yes | 11,260 | (26.2) | 900 | (42.3) | < 0.001 | 8,403 | (25.2) | 3,757 | (32.2) | < 0.001 |
| | No | 31,643 | (73.8) | 1,228 | (57.7) | | 24,964 | (74.8) | 7,907 | (67.8) | |
| Sufficient sleep | | | | | | | | | | | |
| | Yes | 31,367 | (76.7) | 1,590 | (78.6) | 0.043 | 24,180 | (76.3) | 8,777 | (78.0) | < 0.001 |
| | No | 9,549 | (23.3) | 432 | (21.4) | | 7,507 | (23.7) | 2,474 | (22.0) | |

[a]P values are calculated using the chi-square test.

| Diabetes | | | | | Dyslipidemia | | | | |
| --- | --- | --- | --- | --- | --- | --- | --- | --- | --- |
| Undeveloped | | Developed | | P value[a] | Undeveloped | | Developed | | P value[a] |
| n = 46,600 | | n = 1,203 | | | n = 31,930 | | n = 15,873 | | |
| n | (%) | n | (%) | | n | (%) | n | (%) | |
| 17,966 | (38.6) | 685 | (56.9) | **< 0.001** | 12,851 | (40.2) | 5,800 | (36.5) | **< 0.001** |
| 28,634 | (61.4) | 518 | (43.1) | | 19,079 | (59.8) | 10,073 | (63.5) | |
| 3,301 | (7.1) | 34 | (2.8) | **< 0.001** | 2,498 | (7.8) | 837 | (5.3) | **< 0.001** |
| 3,288 | (7.1) | 41 | (3.4) | | 2,287 | (7.2) | 1,042 | (6.6) | |
| 3,114 | (6.7) | 47 | (3.9) | | 2,042 | (6.4) | 1,119 | (7.0) | |
| 4,066 | (8.7) | 93 | (7.7) | | 2,529 | (7.9) | 1,630 | (10.3) | |
| 9,438 | (20.3) | 263 | (21.9) | | 6,055 | (19.0) | 3,646 | (23.0) | |
| 16,589 | (35.6) | 562 | (46.7) | | 11,231 | (35.2) | 5,920 | (37.3) | |
| 6,804 | (14.6) | 163 | (13.5) | | 5,288 | (16.6) | 1,679 | (10.6) | |
| 6,395 | (13.7) | 206 | (17.1) | **0.001** | 4,325 | (13.5) | 2,276 | (14.3) | **0.018** |
| 40,201 | (86.3) | 996 | (82.9) | | 27,602 | (86.5) | 13,595 | (85.7) | |
| 7,929 | (18.9) | 320 | (29.4) | **< 0.001** | 5,083 | (17.7) | 3,166 | (22.0) | **< 0.001** |
| 34,087 | (81.1) | 767 | (70.6) | | 23,635 | (82.3) | 11,219 | (78.0) | |
| 18,826 | (42.2) | 553 | (48.6) | **< 0.001** | 13,056 | (42.8) | 6,323 | (41.6) | **0.023** |
| 25,745 | (57.8) | 584 | (51.4) | | 17,470 | (57.2) | 8,859 | (58.4) | |
| 21,894 | (52.1) | 590 | (54.5) | 0.129 | 15,121 | (52.7) | 7,363 | (51.2) | **0.004** |
| 20,090 | (47.9) | 492 | (45.5) | | 13,574 | (47.3) | 7,008 | (48.8) | |
| 18,689 | (44.8) | 488 | (45.1) | 0.847 | 12,776 | (44.8) | 6,401 | (44.8) | 0.981 |
| 23,066 | (55.2) | 594 | (54.9) | | 15,759 | (55.2) | 7,901 | (55.2) | |
| 32,793 | (78.1) | 800 | (73.5) | **< 0.001** | 22,551 | (78.5) | 11,042 | (76.8) | **< 0.001** |
| 9,214 | (21.9) | 288 | (26.5) | | 6,162 | (21.5) | 3,340 | (23.2) | |
| 6,549 | (15.6) | 167 | (15.4) | 0.892 | 4,586 | (16.0) | 2,130 | (14.8) | **0.002** |
| 35,458 | (84.4) | 918 | (84.6) | | 24,122 | (84.0) | 12,254 | (85.2) | |
| 3,328 | (7.9) | 51 | (4.7) | **< 0.001** | 2,180 | (7.6) | 1,199 | (8.3) | **0.007** |
| 38,665 | (92.1) | 1,035 | (95.3) | | 26,522 | (92.4) | 13,178 | (91.7) | |
| 11,832 | (27.0) | 328 | (28.8) | 0.178 | 8,518 | (28.3) | 3,642 | (24.3) | **< 0.001** |
| 32,060 | (73.0) | 811 | (71.2) | | 21,549 | (71.7) | 11,322 | (75.7) | |
| 32,097 | (76.7) | 860 | (79.5) | **0.034** | 22,043 | (77.1) | 10,914 | (76.2) | **0.041** |
| 9,759 | (23.3) | 222 | (20.5) | | 6,565 | (22.9) | 3,416 | (23.8) | |

**Table 2. Correlation between lifestyle behaviors and lifestyle diseases determined using a logistic regression model.**

| | | Univariate | | | multivariate | | | | | |
| | | | | | model1[a] | | | model2[b] | | |
| Lifestyle behaviors | | OR | 95%CI | P value | OR | 95%CI | P value | OR | 95%CI | P value |
|---|---|---|---|---|---|---|---|---|---|---|
| **A) Metabolic syndrome** | | | | | | | | | | |
| Smoking | No | 1.00 | (reference) | | 1.00 | (reference) | | 1.00 | (reference) | |
| | Yes | **1.92** | (1.74-2.13) | **< 0.001** | **1.38** | (1.22-1.56) | **< 0.001** | **1.50** | (1.32-1.70) | **< 0.001** |
| Weight gain of ≥10 kg since the age of 20 years | No | 1.00 | (reference) | | 1.00 | (reference) | | 1.00 | (reference) | |
| | Yes | **6.40** | (5.85-7.02) | **< 0.001** | **5.46** | (4.95-6.03) | **< 0.001** | **2.01** | (1.79-2.25) | **< 0.001** |
| Regular exercise | Yes | 1.00 | (reference) | | 1.00 | (reference) | | 1.00 | (reference) | |
| | No | 0.93 | (0.86-1.02) | 0.120 | 0.94 | (0.85-1.05) | 0.307 | 0.96 | (0.86-1.08) | 0.505 |
| Physical activity | Yes | 1.00 | (reference) | | 1.00 | (reference) | | 1.00 | (reference) | |
| | No | **1.12** | (1.03-1.23) | **0.010** | 1.05 | (0.94-1.16) | 0.418 | 1.03 | (0.92-1.15) | 0.638 |
| Walking speed | Fast | 1.00 | (reference) | | 1.00 | (reference) | | 1.00 | (reference) | |
| | Slow | **1.20** | (1.10-1.31) | **< 0.001** | **1.25** | (1.13-1.38) | **< 0.001** | **1.16** | (1.05-1.30) | **0.005** |
| Eating speed | Slow/Normal | 1.00 | (reference) | | 1.00 | (reference) | | 1.00 | (reference) | |
| | Fast | **1.65** | (1.50-1.82) | **< 0.001** | **1.27** | (1.15-1.42) | **< 0.001** | 1.10 | (0.99-1.23) | 0.090 |
| Eating before bedtime | No | 1.00 | (reference) | | 1.00 | (reference) | | 1.00 | (reference) | |
| | Yes | **1.42** | (1.27-1.59) | **< 0.001** | 1.01 | (0.90-1.15) | 0.817 | 0.96 | (0.84-1.09) | 0.495 |
| Skipping breakfast | No | 1.00 | (reference) | | 1.00 | (reference) | | 1.00 | (reference) | |
| | Yes | **1.28** | (1.10-1.49) | **0.001** | 0.98 | (0.83-1.17) | 0.859 | 1.05 | (0.87-1.25) | 0.615 |
| Problematic drinking habits | No | 1.00 | (reference) | | 1.00 | (reference) | | 1.00 | (reference) | |
| | Yes | **2.06** | (1.89-2.25) | **< 0.001** | **1.18** | (1.06-1.31) | **0.003** | **1.14** | (1.02-1.28) | **0.024** |
| Sufficient sleep | Yes | 1.00 | (reference) | | 1.00 | (reference) | | 1.00 | (reference) | |
| | No | **0.89** | (0.80-0.99) | **0.040** | 0.93 | (0.83-1.05) | 0.253 | 1.01 | (0.90-1.15) | 0.825 |
| **B) Hypertension** | | | | | | | | | | |
| Smoking | No | 1.00 | (reference) | | 1.00 | (reference) | | 1.00 | (reference) | |
| | Yes | **1.13** | (1.07-1.2) | **< 0.001** | **1.11** | (1.04-1.19) | **0.003** | **1.18** | (1.10-1.27) | **< 0.001** |
| Weight gain of ≥10 kg since the age of 20 years | No | 1.00 | (reference) | | 1.00 | (reference) | | 1.00 | (reference) | |
| | Yes | **1.44** | (1.37-1.52) | **< 0.001** | **1.44** | (1.36-1.53) | **< 0.001** | **1.15** | (1.07-1.23) | **< 0.001** |
| Regular exercise | Yes | 1.00 | (reference) | | 1.00 | (reference) | | 1.00 | (reference) | |
| | No | **0.93** | (0.89-0.97) | **0.001** | **1.06** | (1.01-1.12) | **0.019** | **1.06** | (1.01-1.12) | **0.032** |
| Physical activity | Yes | 1.00 | (reference) | | 1.00 | (reference) | | 1.00 | (reference) | |
| | No | **0.95** | (0.91-0.99) | **0.021** | 0.96 | (0.91-1.00) | 0.073 | 0.96 | (0.91-1.01) | 0.132 |
| Walking speed | Fast | 1.00 | (reference) | | 1.00 | (reference) | | 1.00 | (reference) | |
| | Slow | 1.00 | (0.95-1.04) | 0.832 | **1.06** | (1.01-1.11) | **0.012** | **1.06** | (1.01-1.12) | **0.019** |
| Eating speed | Slow/Normal | 1.00 | (reference) | | 1.00 | (reference) | | 1.00 | (reference) | |
| | Fast | 1.03 | (0.98-1.09) | 0.210 | 1.03 | (0.97-1.08) | 0.341 | 1.00 | (0.94-1.06) | 0.972 |
| Eating before bedtime | No | 1.00 | (reference) | | 1.00 | (reference) | | 1.00 | (reference) | |
| | Yes | **1.11** | (1.05-1.18) | **< 0.001** | **1.08** | (1.01-1.15) | **0.016** | 1.05 | (0.98-1.13) | 0.135 |
| Skipping breakfast | No | 1.00 | (reference) | | 1.00 | (reference) | | 1.00 | (reference) | |
| | Yes | **0.92** | (0.84-0.99) | **0.033** | 1.03 | (0.94-1.13) | 0.541 | 1.04 | (0.94-1.14) | 0.455 |
| Problematic drinking habits | No | 1.00 | (reference) | | 1.00 | (reference) | | 1.00 | (reference) | |
| | Yes | **1.41** | (1.35-1.48) | **< 0.001** | **1.38** | (1.31-1.46) | **< 0.001** | **1.22** | (1.15-1.30) | **< 0.001** |
| Sufficient sleep | Yes | 1.00 | (reference) | | 1.00 | (reference) | | 1.00 | (reference) | |
| | No | **0.91** | (0.86-0.96) | **< 0.001** | 0.96 | (0.91-1.02) | 0.189 | 0.99 | (0.93-1.05) | 0.747 |
| **C) diabetes** | | | | | | | | | | |
| Smoking | No | 1.00 | (reference) | | 1.00 | (reference) | | 1.00 | (reference) | |

*(Continued)*

**Table 2.** (Continued)

| Lifestyle behaviors | | Univariate | | | multivariate model1[a] | | | model2[b] | | |
|---|---|---|---|---|---|---|---|---|---|---|
| | | OR | 95%CI | P value | OR | 95%CI | P value | OR | 95%CI | P value |
| | Yes | **1.30** | (1.12-1.51) | **0.001** | 1.14 | (0.96-1.37) | 0.132 | 1.17 | (0.98-1.41) | 0.086 |
| Weight gain of ≥10 kg since the age of 20 years | No | 1.00 | (reference) | | 1.00 | (reference) | | 1.00 | (reference) | |
| | Yes | **1.79** | (1.57-2.05) | **< 0.001** | **1.62** | (1.41-1.87) | **< 0.001** | 1.11 | (0.94-1.32) | 0.222 |
| Regular exercise | Yes | 1.00 | (reference) | | 1.00 | (reference) | | 1.00 | (reference) | |
| | No | **0.77** | (0.69-0.87) | **< 0.001** | 0.91 | (0.79-1.05) | 0.203 | 0.95 | (0.82-1.10) | 0.489 |
| Physical activity | Yes | 1.00 | (reference) | | 1.00 | (reference) | | 1.00 | (reference) | |
| | No | 0.91 | (0.80-1.03) | 0.122 | 0.97 | (0.85-1.12) | 0.707 | 0.95 | (0.83-1.10) | 0.491 |
| Walking speed | Fast | 1.00 | (reference) | | 1.00 | (reference) | | 1.00 | (reference) | |
| | Slow | 1.01 | (0.90-1.14) | 0.823 | 1.14 | (0.99-1.30) | 0.050 | 1.13 | (0.99-1.30) | 0.075 |
| Eating speed | Slow/Normal | 1.00 | (reference) | | 1.00 | (reference) | | 1.00 | (reference) | |
| | Fast | **1.28** | (1.12-1.47) | **< 0.001** | **1.24** | (1.08-1.44) | **0.003** | **1.17** | (1.01-1.36) | **0.037** |
| Eating before bedtime | No | 1.00 | (reference) | | 1.00 | (reference) | | 1.00 | (reference) | |
| | Yes | 0.98 | (0.83-1.16) | 0.859 | 0.98 | (0.82-1.17) | 0.839 | 0.96 | (0.80-1.15) | 0.639 |
| Skipping breakfast | No | 1.00 | (reference) | | 1.00 | (reference) | | 1.00 | (reference) | |
| | Yes | **0.57** | (0.43-0.76) | **< 0.001** | **0.64** | (0.48-0.87) | **0.004** | **0.70** | (0.51-0.95) | **0.021** |
| Problematic drinking habits | No | 1.00 | (reference) | | 1.00 | (reference) | | 1.00 | (reference) | |
| | Yes | 1.10 | (0.96-1.25) | 0.167 | **0.79** | (0.68-0.92) | **0.002** | 0.94 | (0.81-1.10) | 0.463 |
| Sufficient sleep | Yes | 1.00 | (reference) | | 1.00 | (reference) | | 1.00 | (reference) | |
| | No | **0.85** | (0.73-0.99) | **0.032** | 0.95 | (0.81-1.11) | 0.500 | 0.95 | (0.81-1.12) | 0.547 |
| **D) Dyslipidemia** | | | | | | | | | | |
| Smoking | No | 1.00 | (reference) | | 1.00 | (reference) | | 1.00 | (reference) | |
| | Yes | **1.07** | (1.01-1.13) | **0.018** | **1.16** | (1.09-1.24) | **< 0.001** | **1.26** | (1.17-1.35) | **< 0.001** |
| Weight gain of ≥10 kg since the age of 20 years | No | 1.00 | (reference) | | 1.00 | (reference) | | 1.00 | (reference) | |
| | Yes | **1.31** | (1.25-1.38) | **< 0.001** | **1.35** | (1.28-1.43) | **< 0.001** | **1.17** | (1.10-1.25) | **< 0.001** |
| Regular exercise | Yes | 1.00 | (reference) | | 1.00 | (reference) | | 1.00 | (reference) | |
| | No | **1.05** | (1.01-1.09) | **0.022** | 1.00 | (0.96-1.05) | 0.927 | 1.01 | (0.96-1.06) | 0.784 |
| Physical activity | Yes | 1.00 | (reference) | | 1.00 | (reference) | | 1.00 | (reference) | |
| | No | **1.06** | (1.02-1.10) | **0.004** | 1.02 | (0.97-1.07) | 0.427 | 1.01 | (0.96-1.06) | 0.616 |
| Walking speed | Fast | 1.00 | (reference) | | 1.00 | (reference) | | 1.00 | (reference) | |
| | Slow | 1.00 | (0.96-1.04) | 0.973 | 0.98 | (0.94-1.02) | 0.296 | 0.98 | (0.94-1.03) | 0.431 |
| Eating speed | Slow/Normal | 1.00 | (reference) | | 1.00 | (reference) | | 1.00 | (reference) | |
| | Fast | **1.11** | (1.06-1.16) | **< 0.001** | **1.09** | (1.04-1.15) | **0.001** | 1.05 | (0.99-1.11) | 0.069 |
| Eating before bedtime | No | 1.00 | (reference) | | 1.00 | (reference) | | 1.00 | (reference) | |
| | Yes | **0.91** | (0.87-0.97) | **0.002** | **0.93** | (0.88-0.99) | **0.019** | 0.95 | (0.89-1.01) | 0.125 |
| Skipping breakfast | No | 1.00 | (reference) | | 1.00 | (reference) | | 1.00 | (reference) | |
| | Yes | **1.11** | (1.03-1.19) | **0.007** | **1.13** | (1.04-1.23) | **0.003** | **1.10** | (1.01-1.20) | **0.028** |
| Problematic drinking habits | No | 1.00 | (reference) | | 1.00 | (reference) | | 1.00 | (reference) | |
| | Yes | **0.81** | (0.78-0.85) | **< 0.001** | **0.81** | (0.77-0.86) | **< 0.001** | **0.90** | (0.85-0.95) | **< 0.001** |
| Sufficient sleep | Yes | 1.00 | (reference) | | 1.00 | (reference) | | 1.00 | (reference) | |
| | No | **1.05** | (1.00-1.10) | **0.040** | 1.02 | (0.97-1.07) | 0.504 | **1.06** | (1.00-1.11) | **0.045** |

[a] Model 1 included the 10 lifestyle behaviors as independent variables and was adjusted for sex and age as covariates.

[b] Model 2 was adjusted for Model 1 by adding BMI and laboratory tests as covariates.

our study showed that skipping breakfast protects against the onset of diabetes. Previous studies have reported a positive correlation between skipping breakfast and diabetes onset [15,16], whereas others have reported no correlation [17]. Our research suggests a negative correlation between skipping breakfast and diabetes onset, which contradicts those of previous studies and presents new insights into this issue. A previous study has highlighted the importance of considering both energy intake and dietary content [26]. However, the unique dietary habits in the Gunma Prefecture may have influenced this result. According to the NHNS in 2016, despite similar daily energy intake, the residents of Gunma Prefecture did consume significantly more carbohydrates compared with the national average, potentially affecting glucose metabolism and insulin sensitivity. Additionally, meal timing and overall dietary patterns could play a role. A previous study suggest that prolonged fasting intervals may improve metabolic health by enhancing insulin sensitivity and glycemic control [32]. It is possible that individuals who skip breakfast in this population may compensate with well-balanced meals later in the day, mitigating the adverse effects typically associated with breakfast skipping. Nonetheless, alternative explanations should be considered. For example, reverse causation may partly explain this finding—for instance, individuals with preclinical diabetes may have modified their eating habits by skipping breakfast to manage symptoms or weight. In addition, since dietary habits were self-reported, reporting bias may have influenced the observed association. Furthermore, our study was limited to insured adults within Gunma Prefecture, which may restrict the generalizability of our findings. Finally, unmeasured confounders such as sleep patterns, work schedules, and socioeconomic status could also have affected the results. Taken together, these factors highlight the need for cautious interpretation of our findings and suggest that further studies using objective dietary assessments and longitudinal data are warranted to clarify these associations.

Second, we identified an association between MetS and lifestyle behaviors. Previous studies have reported smoking [33], weight gain ≥10 kg since the age of 20 years [34], slow walking [13], fast eating [35,36], skipping breakfast (for men) [37], problematic drinking habits [13,38,39], and adopting poor sleeping habits (for men) [37] as MetS risk factors. Our study showed that smoking, weight gain ≥10 kg since the age of 20 years, slow walking, and problematic drinking habits were risk factors in Model 2, while five factors were identified in Model 1 (Table 2A). The remaining factor, fast eating, might be associated with measurement tests that influence the onset of MetS and previous study have suggested the need for adjustments based on these test items [10].

Third, we identified an association between hypertension and lifestyle behaviors. Previous studies have reported smoking [23,40], weight gain ≥10 kg since the age of 20 years [34], the lack of regular exercise [26], slow walking [23], fast eating [23], problematic drinking habits [39–41], and adopting poor sleeping habits [23] as hypertension risk factors. Our study showed that smoking, weight gain of ≥10 kg since the age of 20 years, the lack of regular exercise, slow walking, and problematic drinking habits were risk factors in Model 2, while six factors were identified in Model 1 (Table 2B). The remaining factor, eating before bedtime, might be associated with measurement tests that influence the onset of hypertension and previous study have suggested the need for adjustments based on these test items [11].

Fourth, we identified an association between dyslipidemia and lifestyle behaviors. Previous studies have reported smoking [42], weight gain ≥10 kg since the age of 20 years [34], skipping breakfast [43], and adopting poor sleeping habits [44] as dyslipidemia risk factors. These lifestyle behaviors were similarly confirmed as risk factors in Model 2, while six factors were identified in Model 1 (Table 2D). The remaining two factors, fast eating and eating before bedtime might be associated with measurement tests that influence the onset of dyslipidemia and previous studies have suggested the need for adjustments based on these test items [11,36]. Additionally, our study showed that drinking habits are protective against the onset of dyslipidemia, which aligns with previous research finding [39].

Thus, we showed that utilizing the results of self-administered questionnaires from SHC is effective in preventing lifestyle diseases. Notably, smoking and weight gain ≥10 kg since the age of 20 years were identified as common risk factors for MetS, hypertension, and dyslipidemia. The findings of this study provide essential evidence to support and strengthen targeted public health interventions, such as smoking cessation programs and weight management strategies based on

the Health Promotion Act. Addressing these two factors through such interventions could play a crucial role in disease prevention and risk reduction.

The impact of lifestyle factors such as smoking, weight management, physical activity, and alcohol consumption on cardiovascular disease and life expectancy has been widely recognized across different populations [45,46]. However, cultural and dietary differences may influence these associations. Our study, which adjusted for 10 detailed lifestyle habits, offers a broader perspective than previous studies. Further research in diverse populations is needed to confirm these findings.

This study has several limitations. First, we did not account for any changes in lifestyle behaviors among individuals who underwent SHCs during the 4-year period following the initial health checkup. Longitudinal studies often struggle to incorporate the changes in background factors into their models; therefore, future research incorporating longitudinal tracking of lifestyle behaviors to account for these changes is necessary. Second, the lifestyle assessment data collected using self-administered questionnaires lack objectivity and are subject to potential biases, such as recall bias and social desirability bias. However, this method has been validated in studies by Takeuchi *et al*. [47] and Fukasawa *et al*. [19], demonstrating its statistical validity. To mitigate potential biases, future studies should consider complementing self-reported data with objective measures, such as wearable devices for physical activity monitoring. Third, the database used comprised information of individuals insured under the Gunma Prefecture's NHI. Consequently, it remains unclear whether these findings can be generalized to other regions or insurance groups. Regional characteristics, economic indicators, and the social determinants of health may influence the outcomes. Future studies in diverse populations and healthcare systems are needed to validate these findings and ensure broader applicability.

In conclusion, this study elucidated the time-line changes associations between lifestyle behaviors and MetS, hypertension, diabetes, and dyslipidemia using SHC data accumulated over 10 years. Additionally, the correlations between lifestyle-related diseases and specific lifestyle habits were evaluated. These results suggest that self-administered questionnaires assessing lifestyle behaviors are useful for predicting future health issues. Notably, smoking and weight gain emerged as common risk factors for these four diseases, highlighting the need for targeted interventions to address these issues.

## Supporting information

**S1 Table. Cutoff values.**
(XLSX)

**S2 Table. List of "standard questionnaire".**
(XLSX)

**S3 Table. BMI, and laboratory values of individuals who developed and did not develop lifestyle diseases.** Supplemental data for Table 1.
(XLSX)

**S1 Fig. Correlation between lifestyle behaviors and lifestyle diseases determined using a multivariable logistic regression model (Supplementary to Table 2).** The odds ratio (black bars) with corresponding 95% confidence intervals (error bars) is shown for each category of lifestyle diseases. *$P<0.05$, **$P<0.01$, ***$P<0.001$ compared with the reference category.
(TIFF)

## Acknowledgments

We express our gratitude to the Gunma Prefecture Health and Welfare Department, Health and Longevity Society Promotion Section, for their invaluable support in facilitating the use of the Gunma Prefecture National Health Insurance data. We would like to thank Editage (www.editage.jp) for English language editing.

## Author contributions

**Investigation:** Takafumi Okawa, Hikaru Negishi.

**Writing – original draft:** Takafumi Okawa.

**Writing – review & editing:** Yuki Aoki, Mitsuo Uchida, Yumi Sato, Mai Ishikawa, Rie Matsui, Kaori Hotta, Takayuki Saitoh.

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
