## [Decision Letter · Decision Letter 0]

PONE-D-24-56824Impact of Lifestyle Behaviors on the Development of Lifestyle Diseases: A Retrospective Cohort StudyPLOS ONE

Dear Dr. Saitoh,

Thank you for submitting your manuscript to PLOS ONE. After careful consideration, we feel that it has merit but does not fully meet PLOS ONE’s publication criteria as it currently stands. Therefore, we invite you to submit a revised version of the manuscript that addresses the points raised during the review process.

We look forward to receiving your revised manuscript.

Kind regards,

Mulu Tiruneh

Academic Editor

PLOS ONE

Reviewers' comments:

Reviewer's Responses to Questions

**Comments to the Author**

1. Is the manuscript technically sound, and do the data support the conclusions?

Reviewer #1: Yes

Reviewer #2: Yes

2. Has the statistical analysis been performed appropriately and rigorously? 

Reviewer #1: Yes

Reviewer #2: Yes

3. Have the authors made all data underlying the findings in their manuscript fully available?

Reviewer #1: Yes

Reviewer #2: Yes

4. Is the manuscript presented in an intelligible fashion and written in standard English?

Reviewer #1: Yes

Reviewer #2: Yes

5. Review Comments to the Author

Reviewer #1: The manuscript presents a comprehensive and well-executed study on the impact of lifestyle behaviors on the development of lifestyle diseases using a robust dataset from a large retrospective cohort. The research is technically sound, with an appropriately designed methodology and statistical analysis that strengthens the reliability of the findings. By utilizing logistic regression models with stepwise adjustments, the study effectively accounts for key confounders, including age, sex, BMI, and laboratory parameters. The large sample size of over 47,000 participants ensures sufficient statistical power, and the stratified analysis of disease outcomes enhances the depth of the results.

The statistical analysis is rigorous, employing multivariate models that explore the independent effects of lifestyle behaviors. However, additional details on multicollinearity diagnostics, such as reporting variance inflation factors (VIFs), would further validate the results. While the use of odds ratios with confidence intervals is appropriate, the discussion could benefit from emphasizing the clinical relevance of statistically significant findings. For instance, while smoking and weight gain emerged as common risk factors across multiple diseases, it would be helpful to contextualize these findings within public health interventions.

One intriguing finding is the protective association between skipping breakfast and diabetes, which contradicts much of the existing literature. While this result is novel, it warrants further exploration, possibly through sensitivity or subgroup analyses to account for regional dietary habits or meal composition. The manuscript would also benefit from a discussion on residual confounding, as factors like socioeconomic status or genetic predispositions were not included in the analysis.

Although the reliance on self-reported data is acknowledged as a limitation, it introduces potential biases such as recall and social desirability bias. A more detailed discussion on how these biases may have influenced the results would strengthen the manuscript. Additionally, the study does not account for changes in lifestyle behaviors during the follow-up period, which may affect the observed associations. Future research should consider longitudinal tracking of behavioral changes to provide a more nuanced understanding of their effects.

In terms of generalizability, the study is limited to a specific region in Japan, and the findings may not be directly applicable to other populations with different cultural, dietary, or healthcare contexts. The authors should discuss how similar studies in other regions could validate and expand upon these results.

In conclusion, this manuscript is a valuable contribution to the field of public health and provides actionable insights into the relationship between lifestyle behaviors and lifestyle diseases. With minor improvements in the discussion of limitations, clinical relevance, and generalizability, this study could serve as a strong foundation for public health strategies aimed at mitigating the burden of lifestyle diseases.

Kudos to authors and team member in developing a very good project and article.

Reviewer #2: Strengths:

The study addresses a critical public health issue based on a large longitudinal dataset.

The application of the logistic regression model is appropriate, and the results are presented clearly.

The supplementary materials included herein enhance the reproducibility of the study.

Suggestions for Improvement:

Explain why the analysis excluded individuals who had multiple checkups. If this is done, selection bias could be introduced into the analysis because such persons may differ in some systematic way from those included in the analysis.

Address the multicollinearity issue in the logistic regression models by reporting the VIF values and discussing how the overlapping lifestyle behaviors were dealt with.

Simplify and visualize results using plots of bar charts or heat maps that give in one glance several key findings of Table 1 and Table 2.

The findings should be discussed for their clinical significance. However, not every statistically significant association is clinically significant and may deserve further consideration in terms of modest effect size, such as the identified one for fast eating and diabetes, with an OR of 1.17. Expand the discussion of limitations. Acknowledge potential biases (e.g., recall bias, selection bias) and the limited generalizability of the findings.

Highlight practical implications: Describe how such findings might be used to inform public health interventions or policy.

Additional Comments:

The negative relation of breakfast omission with diabetes appears intriguingly protective, with an odds ratio of 0.70, which also goes against previous reports. This might be discussed more elaborately in the discussion part, considering either regional dietary habits or confounding. Where relevant, for example, when referring to the "standard questionnaire" or cutoff values, the manuscript should refer explicitly to supplementary tables.

6. PLOS authors have the option to publish the peer review history of their article (what does this mean? ). If published, this will include your full peer review and any attached files.

**Do you want your identity to be public for this peer review?** For information about this choice, including consent withdrawal, please see our Privacy Policy .

Reviewer #1: No

Reviewer #2: **Yes: ** Kola Adegoke

---

## [Author Response · Author response to Decision Letter 1]

17 Apr 2025

We appreciate the constructive feedback from the reviewers and the editor. We have carefully revised the manuscript in response to their comments. A detailed point-by-point response is provided in the attached document.

---

## [Decision Letter · Decision Letter 1]

PONE-D-24-56824R1Impact of Lifestyle Behaviors on the Development of Lifestyle Diseases: A Retrospective Cohort StudyPLOS ONE

Dear Dr. Saitoh,

Thank you for submitting your manuscript to PLOS ONE. After careful consideration, we feel that it has merit but does not fully meet PLOS ONE’s publication criteria as it currently stands. Therefore, we invite you to submit a revised version of the manuscript that addresses the points raised during the review process.

We look forward to receiving your revised manuscript.

Kind regards,

Mulu Tiruneh

Academic Editor

PLOS ONE

Journal Requirements:

Reviewers' comments:

Reviewer's Responses to Questions

**Comments to the Author**

1. If the authors have adequately addressed your comments raised in a previous round of review and you feel that this manuscript is now acceptable for publication, you may indicate that here to bypass the “Comments to the Author” section, enter your conflict of interest statement in the “Confidential to Editor” section, and submit your "Accept" recommendation.

Reviewer #2: (No Response)

2. Is the manuscript technically sound, and do the data support the conclusions?

Reviewer #2: Partly

3. Has the statistical analysis been performed appropriately and rigorously? 

Reviewer #2: Yes

4. Have the authors made all data underlying the findings in their manuscript fully available?

Reviewer #2: No

5. Is the manuscript presented in an intelligible fashion and written in standard English?

Reviewer #2: Yes

6. Review Comments to the Author

Reviewer #2: We appreciate the chance to comment on the revised manuscript entitled "Impact of Lifestyle Behavior on Lifestyle Disease Development: A Retrospective Cohort Study." The study is well-timed and methodologically rigorous, using a large dataset to assess self-reported lifestyle behavior predictors for the development of metabolic syndrome, diabetes, hypertension, and dyslipidemia. The statistical strategy, which involves logistic regression models and stepwise adjustments, is suitable and well-explained. The authors enhance the manuscript further through a more explicit elaboration of rationale and substantiation for the relationship between behavioral surveillance and prediction of public health risk.

However, I want to voice my concern over the findings' interpretability and generalizability. The implication that skipping breakfast is linked to less diabetes contradicts the literature. It suggests a more elaborate sensitivity analysis or other explanations, such as reverse causation or bias for reporting. Although the data are extensive, the sample is geographically and demographically restricted (only insured adults within Gunma Prefecture), and this may be better highlighted. The confounding effect from unmeasured lifestyles or socioeconomic factors (e.g., sleep, work, job status) could also be mentioned.

Finally, whilst the manuscript is composed well and readable, the Data Availability Statement is inconsistent with PLOS’s open data policy. Please specify a means for de-identified data availability through a recognized repository or institutional arrangements. I suggest minor revisions and changes to meet these issues and enhance the manuscript's contribution to the evidence base for preventive health actions through behavioral data.

7. PLOS authors have the option to publish the peer review history of their article (what does this mean? ). If published, this will include your full peer review and any attached files.

**Do you want your identity to be public for this peer review?** For information about this choice, including consent withdrawal, please see our Privacy Policy .

Reviewer #2: **Yes: ** Kola Adegoke

---

## [Author Response · Author response to Decision Letter 2]

11 Jun 2025

We thank the editor and reviewers for their thoughtful and constructive comments. We have carefully revised the manuscript in response to all points raised. A detailed point-by-point response is provided in the attached response letter.

---

## [Editor Report · Decision Letter 2]

Impact of Lifestyle Behaviors on the Development of Lifestyle Diseases: A Retrospective Cohort Study

PONE-D-24-56824R2

Dear Dr. Saitoh,

We’re pleased to inform you that your manuscript has been judged scientifically suitable for publication and will be formally accepted for publication once it meets all outstanding technical requirements.

Kind regards,

Mulu Tiruneh

Academic Editor

PLOS ONE
---

## [Editor Report · Acceptance letter]

PONE-D-24-56824R2

PLOS ONE

Dear Dr. Saitoh,

I'm pleased to inform you that your manuscript has been deemed suitable for publication in PLOS ONE. Congratulations! Your manuscript is now being handed over to our production team.

Kind regards,

on behalf of

Mr. Mulu Tiruneh

Academic Editor

PLOS ONE